# Benign Fasciculation Syndrome and Migraine Aura without Headache: Possible Rare Side Effects of the BNT162b2 mRNA Vaccine? A Case Report and a Potential Hypothesis

**DOI:** 10.3390/vaccines10010117

**Published:** 2022-01-13

**Authors:** Grgur Salai, Ervina Bilic, Dragan Primorac, Darija Mahovic Lakusic, Hrvoje Bilic, Ines Lazibat, Lovorka Grgurevic

**Affiliations:** 1Center for Translational and Clinical Research, Department of Proteomics, School of Medicine, University of Zagreb, 10000 Zagreb, Croatia; salai.grgur@gmail.com; 2Teaching Institute of Emergency Medicine of the City of Zagreb, 10000 Zagreb, Croatia; 3Department of Neurology, University Hospital Centre Zagreb, 10000 Zagreb, Croatia; ervina.bilic@mef.hr (E.B.); darija.mahovic@mef.hr (D.M.L.); hbilic45@gmail.com (H.B.); 4School of Medicine, University of Zagreb, 10000 Zagreb, Croatia; 5St. Catherine Specialty Hospital, 10000 Zagreb, Croatia; draganprimorac2@gmail.com; 6Faculty of Dental Medicine and Health, Josip Juraj Strossmayer University of Osijek, 31000 Osijek, Croatia; ilazibat@kbd.hr; 7Eberly College of Science, The Pennsylvania State University, University Park, PA 16802, USA; 8The Henry C. Lee College of Criminal Justice and Forensic Sciences, University of New Haven, West Haven, CT 06516, USA; 9Medical School, University of Split, 21000 Split, Croatia; 10School of Medicine, Josip Juraj Strossmayer University of Osijek, 31000 Osijek, Croatia; 11Medical School, University of Rijeka, 51000 Rijeka, Croatia; 12Medical School REGIOMED, 96450 Coburg, Germany; 13Department of Neurology, Clinical Hospital Dubrava, 10000 Zagreb, Croatia

**Keywords:** benign fasciculation syndrome, migraine aura without headache, COVID-19 vaccine, BNT162 vaccine, Comirnaty

## Abstract

The BNT162b2 (Pfizer BioNTech) mRNA vaccine is an effective vaccine against COVID-19 infection. Here, we report an adverse event following immunization (AEFI) in a 48-year-old female patient who presented with fasciculations, migraine auras without headaches and in an increased discomfort of previously present palpitations, as well as excitation and insomnia. Her fasciculations were intermittently present until the time this paper was written, starting from the 6th day post-vaccination; they changed localization and frequency, but most commonly they were generalized, affecting almost all muscle groups. The patient also suffered from two incidents of migraine auras with visual kaleidoscope-like phenomena without headaches a few months after the vaccination. These symptoms were considered to be AEFI and no causal relation with the vaccine could be proven.

## 1. Introduction

The development of the COVID-19 vaccine is a major breakthrough in fighting, containing and ending the global pandemic. The BNT162b2 (Pfizer BioNTech; Comirnaty) mRNA vaccine has proven to effectively save lives and decrease the spread of the SARS-CoV-2 virus [1]. An adverse event following immunization (AEFI) is defined as any untoward medical occurrence which follows immunization and does not necessarily have a causal link with the vaccine [2,3]. New information regarding COVID-19 infection and SARS-CoV-2 is being uncovered every day. Thus far, several neurological AEFIs related to the BNT162b2 vaccine have been reported, some examples of which are Bell’s palsy, Guillain Barre syndrome, multiple sclerosis-like central nervous system demyelination syndrome and rhombencephalitis, as well as functional neurological disorders. The pathophysiological mechanisms for these phenomena in potential relation to the vaccine are still mostly unknown [4,5,6,7,8,9,10,11]. Luckily, preliminary data indicate that neurological complications from SARS-CoV-2 vaccines are rare in incidence [12]. Here, we present the case of a patient with AEFI whose symptoms, that were benign in nature, developed shortly after receiving the first dose of the BNT162b2 vaccine. They included muscle fasciculations (muscle twitching) and migraine aura without headache, as well as palpitations, excitation and insomnia.

Fasciculations are spontaneous, intermittently occurring, fast contractions of muscle fibers which are visible to the naked eye [13]. Benign fasciculations are common in the general population; they are localized to a single muscle and are neither multifocal, nor progressive in nature [14]. Typical aura without headache (or migraine aura without headache, “silent migraine”) is deemed to be, by the International Headache Society, a “recurrent disorder manifesting in attacks of reversible focal neurological symptoms that usually develop gradually over 5–20 min and last for less than 60 min” that is “neither accompanied nor followed by a headache of any sort” [15,16]. To our knowledge, this is the first case reported of the syndrome of benign fasciculations and migraine aura without headache that developed after COVID-19 vaccination.

## 2. Case Report

We present a 48-year-old Caucasian female patient with regular menstrual cycles (30–32 days) suffering from benign fasciculation syndrome that began shortly after receiving the first dose of the BNT162b2 mRNA vaccine. Here, we present the course of her symptoms during the period from 12 January 2021 to the time when the paper was written (Figure 1), which, due to the time proximity, might be related with the vaccine. To her knowledge, she has not been infected with or actively exposed to the SARS-CoV-2 virus.

Our patient received the first dose of the BNT162b2 (batch No. 1484) mRNA vaccine on 12th of January; the vaccine was intramuscularly administered in her left deltoid muscle. Two days post-vaccination, she experienced a feeling of hyperexcitation which lasted several days, as well as muscle soreness and increased fatigue during usual daily activities. These symptoms were followed by insomnia which continued for two months. Six days post-vaccination she noticed palpebral twitching of her right lower eyelid. In subsequent days, these fasciculations progressed to the entire body (generalized), first affecting the upper limbs (most common locus being in her left shoulder), but, after several days, they spread to all muscle groups, changing frequency and localization. The patient led a fasciculation diary from which she observed that they were most frequently felt at periods of rest; no connection with the lack of sleep, physical or emotional stress, caffeine or menstrual cycle was found. The frequency, intensity and localization of fasciculations all appeared to change arbitrarily and no potential exacerbation factor was observed. The patient had been taking 2.5 mg bisoprolol daily for the past ten years in order to control palpitations caused by supraventricular extrasystoles (SVES) and had a mild microcytic syderopenic anemia due to menstrual blood loss; she was otherwise healthy, had regular menstrual cycle, normal body mass index and had never before experienced fasciculations, migraine headaches, nor auras. Following a neurological examination, the neurologist confirmed the presence of fasciculations and the patient had a positive ischemia-induced and marginally positive hyperventilation-induced latent tetany test. Furthermore, nerve conduction studies (NCS) and electromyography (EMG) testing were unremarkable. Her serum electrolytes (sodium, potassium, calcium, magnesium, chlorides and phosphates) were all within normal range and she tested negative for anti-gangliozide (anti-GM) antibodies. However, her serum vitamin D levels were marginally decreased. Since then, she had been taking 25,000 i.u. of cholecalciferole weekly. Additionally, shortly after vaccination, she noticed an increase in palpitations and her continuous 24 h electrocardiography (EKG) monitoring showed more than 1000 SVES, so her bisoprolol dosage was increased to a total of 3.75 mg per day in order to successfully control the discomfort of palpitations (Figure 1).

On 19th May (127 days post vaccination), she experienced the first episode of kaleidoscope-like effects in her vision field which lasted for approximately 20 min. These changes started in the center of the vision field, then gradually progressed and, at one point, covered the entire field of vision. The patient presented to the emergency department and was subsequently hospitalized for observation and a diagnostic workup. General, neurological and ophthalmological examinations were unremarkable. Complete blood count showed mild anemia (hemoglobin levels were 110 g/L); serum electrolytes were within the normal range (Table 1). The levels of vitamin D were at the lowest border of the referent range, while PTH levels were normal (Table 1). Creatine kinase levels (that were ordered because of muscle fasciculations) were also normal (78 U/L). Electroencephalography (EEG) showed mild dysrhythmic changes in the right centroparietotemporal region (multi-voltage slow waves with a frequency of 6–7 Hz and up to 50 mV of amplitude). The brain magnetic resonance (MR) and MR angiography were unremarkable. Visual evoked potential (VEP) testing (checkerboard type) was also performed and no conductance abnormalities were found, but a neuronal lesion of the left eye which affected the peripheral parts of the vision field was registered. Color Doppler flow imaging (CDFI) of the vertebral and carotid arteries was also performed and was normal but revealed multiple SVES and thyroid hyperperfusion as accidental findings. The thyroid function tests were within normal range, but antibody titers against thyroglobulin and thyroid peroxidase were elevated (Table 1). A board-certified neurologist concluded that this episode is likely to be aura without headache and recommended the patient to take the following supplements: 25,000 i.u. of cholecalciferole per week and Mg, 500 mg; Ca, 300 mg; vitamin B12, 0.2 μg, daily. The patient had never before experienced migraine-associated phenomena and her family history was also negative for migraine headaches or auras. Since June, she had been feeling pain in her left shoulder which then spread to the site of the deltoid muscle origin and worsened at rest and at night. An ultrasound, on the 25th of August, showed effusion of the tendon sheath of the long head of the bicep muscle and a small effusion of the subacromial–subdeltoid bursa. On the 11th of September, another episode of migraine aura without headaches very similar to the first one occurred. The patient observed that she had decreased her oral magnesium supplementation dose several days prior to this incident. During each visit to the neurologist, the patient was advised not to take the second dose of the vaccine, while her anti-SARS-CoV-2 IgG titer (which she determined independently at several occasions, Figure 2) was still relatively high and offered some protection, due to the fact that the connection between the vaccine and her symptoms was not known and it was deemed that the risks of potential side-effects which might progress with the second dose of the vaccine outweighed the personal risk of COVID-19 mortality for this patient. Furthermore, the patient independently decided to try to quantify her specific T-cell response with the novel interferon-γ release assay (IGRA) for the detection of SARS-CoV-2 specific T-cell response [17]; the IGRA resulted in the value of 1441 mIU/mL (reference range: positive if >200 mIU/mL).

## 3. Discussion

Vaccination against SARS-CoV-2 is the best and proven way of fighting the global pandemic and in saving lives from COVID-19 infection. Here, we present the case of a patient who developed benign fasciculation syndrome (BFS) whose locations changed; thus far, the patient suffered from two episodes of migraine auras (without headache)—these symptoms seemed to be benign in nature, but nevertheless impacted the quality of life.

Generally speaking, the source of fasciculations in BFS still remains a source of debate [18]. Although the precise origin of the fasciculations in BFS cannot be located within a certain compartment of the motor nervous system, it is thought that BFS develops as a consequence of lower motor neuron (LMN) dysfunction. Studies have shown that the ectopic activity in BFS is most frequently located proximal to the distal axonal branching point. On the contrary, in amyotrophic lateral sclerosis (ALS), the source of fasciculations is a complex upper motor neuron dysfunction combined with LMN hyperexcitability [19]. The key clinical difference between fasciculations seen in BFS, compared to those with ALS is that there are no signs of muscle weakness, wasting and brisk tendon reflexes in BFS. New follow-up studies involving patients with BFS have shown that BFS patients never progress to ALS but confirm a chronicity of BFS—the same was seen in the case we present here [20]. Thus, BFS as a clinical and electrophysiologic entity does not necessarily define a place in either the central or peripheral nervous system as a potential site of damage or dysfunction. Moreover, some authors consider BFS of myotomic distribution as a form of limited anterior horn disease and, in some cases, it may be a mild Guillain Barre presentation. BFS after vaccination, together with already well-known neuropathic pain, may point to possible (sensory or motor) neuronopathy as a possible vaccination side effect.

Even though it is by no means possible to claim causality, especially considering the fact that this case is currently fairly isolated in medical literature, due to time proximity to the vaccination event (Figure 1) and the fact that they did not previously occur in this patient, the possibility that these phenomena might be connected with the vaccine should be considered and we currently deem them to be AEFIs. Therefore, we decided to present this case to the scientific and medical community in order to lay a potential foundation stone, i.e., in order to help medical professionals whose patients might have similar ailments that they suspect to be in possible relation to vaccination against SARS-CoV-2.

Muscle twitching is a known neuromuscular manifestation of COVID-19 infection [21]; migraine headaches and auras have been linked with COVID-19 infection in earlier literature, where it was hypothesized that coronaviruses may affect the bioelectrical activity of the brain, especially of the occipital lobes [22]. Furthermore, a case of a 38-year-old female patient with known history of migraines that developed a status of migrainosus one day following the same vaccination has been reported by Consoli et al. [23]. We hypothesize that it is possible this might have been an acute reaction in a “susceptible” patient, whereas a migraine aura without headache in our patient 127 days following vaccination might relate to a subacute autoimmune reaction in a patient unburdened with migraine-related issues. It is worthy to note that fasciculations and migraine-related phenomena have been reported as AEFIs with other vaccines [2,24]. To add, a case of subacromial–subdeltoid bursitis following a different COVID-19 (Oxford—AstraZeneca) vaccine has been reported in the medical literature [25].

Even though our patient had normal thyroid function tests (TFTs), it is important to comment on the fact that she had positive antibodies directed against her thyroid. It cannot be excluded that some of our patient’s symptoms, at least in part, might have been caused by thyroid dysregulation. However, what we observed were normal TFTs at the time of aura (Table 1), as well as normal TFTs in both prior (obtained yearly, during routine check-ups) and subsequent (obtained twice after the first episode of auras) testing. Positive antithyroid antibodies were observed only during the first episode of aura. In the subsequent test (in September), they were negative. We believe that the anti-thyroid antibodies might have been transiently increased during that time, possibly as a result of a presumed “widespread” autoimmune reaction. It is important to note that autoimmune thyroid disease (especially Graves’ disease) has been observed both in patients following COVID-19 infection and in several patients following SARS-CoV-2 vaccination [26,27,28,29].

We hypothesize that, if the presented phenomena were causally connected to the vaccine, that they were mediated by an unknown immunological reaction (potentially involving cross-reactivity and molecular mimicry related to spike proteins). It is possible that these immunological processes attack synapses and neuromuscular junctions at an unknown site (ion-channel?) causing the lowering of the action potential threshold, thus leading to fasciculations, episodes of aura and an increase in SVES. Furthermore, considering the patient’s observation regarding the decrease in magnesium supplementation dose prior to the second episode of aura, this hypothesis is emphasized; magnesium ions are known regulators of membrane potential and have also an emerging role in migraine prophylaxis [30,31]. Autoimmune channelopathies are a group of disorders which are associated with autoantibodies directed against ion channels. It is also possible that spike proteins caused a more generalized immune response similar to that seen in post-COVID-19 syndrome [32,33,34]. Such a “widespread” immune reaction might be revealed in the presence of anti-thyroid antibodies that were found in our patient’s blood sample (in addition to the theorized anti-ion-channel antibodies). Furthermore, the idea that such spike protein-related immune reactions as possibly seen here and in post-COVID-19 syndrome might affect the most sensitive and already micro-damaged organ systems (*locus minoris resistentiae*) is also noteworthy [34]. The mentioned hypotheses are, at this point, of course, mere theories and are impossible to prove.

## 4. Conclusions

We would like to emphasize that the benign fasciculation syndrome and typical aura without headache, in this case, are AEFIs and not proven side effects of vaccination; we do not claim causality. In order to discern whether the causal connection of these symptoms to the vaccine exists, more than a single case will need to be reported and thorough clinical as well as preclinical investigations will need to be performed.

## Figures and Tables

**Figure 1 vaccines-10-00117-f001:**
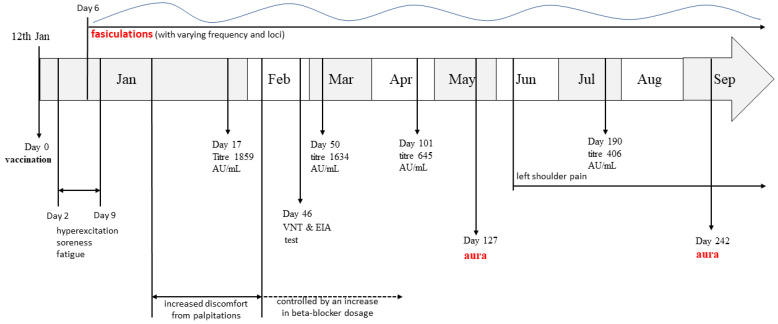
Timeline of the disease course from the point of vaccination (day 0) to the end of September. Important events are timed using days post vaccination. Jan—January; Feb—February; Mar—March; Apr—April; Jun—June; Jul—July; Aug—August; Sep—September; VNT—virus neutralization test; EIA—enzyme immunoassay.

**Figure 2 vaccines-10-00117-f002:**
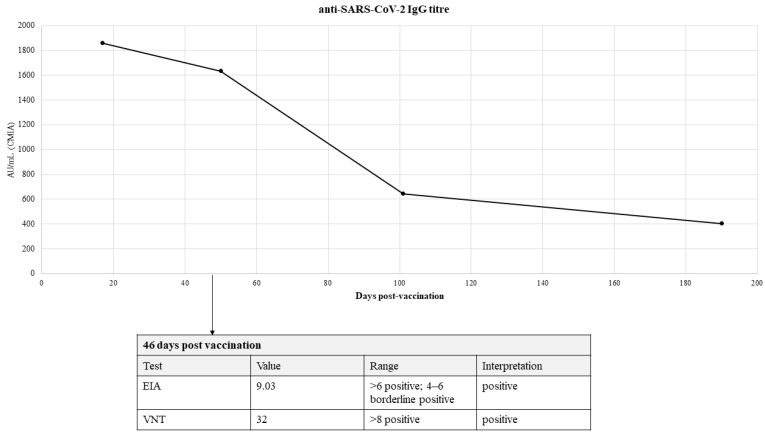
Anti-SARS-CoV-2 IgG titer determined in a private medical laboratory based on a test using chemiluminescent microparticle immunoassay (CMIA) technology. Cut-off value: positive if titer is higher than 50 AU/mL. Enzyme immunoassay (EIA) test for anti-SARS-CoV-2 titer expressed as AI/mL and virus neutralization test (VNT) for SARS-CoV-2 were performed 46 days post-vaccination at the Croatian Institute of Public Health laboratory.

**Table 1 vaccines-10-00117-t001:** Laboratory result summary which includes thyroid function tests and antibody titers against thyroglobulin (anti-Tg) and thyroid peroxidase (anti-TPO), as well as parathyroid hormone (PTH), vitamin D and serum electrolyte statuses. Blood was drawn on 10 June 2021.

149 Days Post Vaccination		
Test	Value	Reference Range
Thyroid stimulating hormone	2.43	0.35–4.9
T4	103	62–150
Anti-Tg	33.5	Positive > 4.1
Anti-TPO	10.05	Positive > 5.61
PTH	4.84	1.6–7.2
25-hidroxy vitamin D	74	Normal > 75 nmol/L
Sodium	139	137–146 mmol/L
Potassium	4.3	3.9–5.1 mmol/L
Ionic calcium	1.21	1.11–1.32 mmol/L
Total magnesium	0.91	0.65–1.05 mmol/L
Inorganic phosphorous	1.25	0.79–1.42 mmol/L

## Data Availability

This case report did not report raw data that is not already presented in the manuscript.

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
