# Peer review of "Benign Fasciculation Syndrome and Migraine Aura without Headache: Possible Rare Side Effects of the BNT162b2 mRNA Vaccine? A Case Report and a Potential Hypothesis"

_vaccines, 2022, doi:10.3390/vaccines10010117_

Round 1

Reviewer 1 Report

Thanks to the authors tp present this case.

It was interesting however some matters exists;

1-introduction should be improved please  explain a little bite the neurological symptoms of covid-19 vaccines.

2-The 127 days distance time from vaccination to aura  is so long to relate it to  the vaccine as the authors wrote.The state of thyroid function should be more precisely monitored since it could explain  all of the symptoms including paplpitation,insomnia,fasciculations and even migraine like aura episodes however the autoimmunity against some location in NMJs is also might be a possibility of the symptoms.

3-I assume the presentation of this case has no a high scientific impact because of the vague etiologies, and variable symptoms in a wide time period.

4-In spite of the low scientific impact i agree to publish this case because of interesting sequential symptoms which all could be created from a nerological anatomic site,i.e,NMJ in CNS synapses and NMJ in LMN area.

Author Response

We wish to thank  Referee 1 for providing us with insightful and important suggestions and remarks, as well as for pointing out the omissions we had made along the way of writing our manuscript. We believe, they truly helped us improve the quality of our manuscript and to improve the reader-experience – to offer future readers interesting and useful information regarding the possible adverse event following immunization with BNT162b2 mRNA vaccine which they may encounter in their clinical practice.

We have rewritten parts of our discussion, as well as updated introduction and case report sections with appropriate information. We used „track changes“ in our manuscript to highlight the changes we made. Furthermore, we stated the most important changes that in our point-by-point response to the Referees (we excluded the reference numbering in the point-by-point response).

Referee: 1

Comments to the Author

Thanks to the authors tp present this case.

We thank the expert Referee for their detailed and precise points and remarks which helped us improve the quality of our paper and for pointing out omissions we had made along the way of creating our manuscript.

It was interesting however some matters exists;

  1. Introduction should be improved please explain a little bite the neurological symptoms of covid-19 vaccines.

We thank the Referee for indicating this omission and helping us improve the quality of our manuscript. We have added the following section in the introduction: “New information regarding COVID-19 infection and SARS-CoV-2 is being uncovered every day. Thus far, several neurological AEFIs related to BNT162b2 vaccine have been reported, some examples of which are Bell's palsy, Guillain Barre syndrome, multiple sclerosis-like central nervous system demyelination syndrome, rhombencefalitis as well as functional neurological disorders. The pathophysiological mechanisms for these phenomena in potential relation to the vaccine are still mostly unknown. Luckily, preliminary data indicate that neurological complications from SARS-CoV-2 vaccines are rare in incidence

  1. The 127 days distance time from vaccination to aura is so long to relate it to the vaccine as the authors wrote. The state of thyroid function should be more precisely monitored since it could explain all of the symptoms including paplpitation, insomnia, fasciculations and even migraine like aura episodes however the autoimmunity against some location in NMJs is also might be a possibility of the symptoms.

We fully agree with the Referee, which is why we claim no causality.  

In terms of hypothesizing, 127 days might relate to a subacute autoimmune manifestation. We would like to briefly mention the report of a possibly “acute” migraine-related AEFI in a previously “susceptible” 38-year old female patient with known history of migraines, that suffered from status migrainosus that originated one day following vaccination (Consoli et al., 2021, Neurol Sci. 2021 Nov 22:1–4).

Regarding the thyroid, what we have observed are normal thyroid function tests (TFT) at the time of aura (Table 1.), as well as normal TFTs in both prior (obtained yearly during routine check-ups) and subsequent (obtained twice after the first episode of aura) testing. Positive antithyroid antibodies were observed only during the first episode of aura. In the subsequent test, they were negative. We believe they might have been transiently increased during that time. It is also possible that they were increased as a result of a “widespread” autoimmune reaction, which was previously hinted in the discussion. We would also like to mention a few recently published cases which report autoimmune thyroid disease in relation to SARS-CoV-2 vaccination: a) Weintraub et. al, 2021, Journal of Investigative Medicine, High Impact Case Reports, Volume 9: 1–4; b) Vera-Lastra et. al, 2021, ThyroidVol. 31, No. 9; c) Lui et al, 2021, Front. Public Health.

Prompted by the Referee’s interesting remarks, we have rewritten part of the discussion related to their remarks: “Muscle twitching is a known neuromuscular manifestations of COVID-19 infection; migraine headaches and auras have been linked with COVID-19 infection in earlier literature where it was hypothesized that coronaviruses may affect bioelectrical activity of the brain, especially of the occipital lobes. Furthermore, a case of a 38 year-old female patient with known history of migraines that developed status migrainosus one day following the same vaccination has been reported by Consoli et al. We hypothesize that it is possible this might be an acute reaction in a „susceptible“ patient, whereas a migraine aura without headache in our patient 127 days following vaccination might relate to a subacute autoimmune reaction in a patient unburdened with migraine-related issues. It is worthy to note that fasciculations and migraine-related phenomena have been reported as AEFI with other vaccines. To add, a case of subacromial-subdeltoid bursitis following a different COVID-19 (Oxford – AstraZeneca) vaccine has been reported in the medical literature..

Even though, our patient had normal thyroid function tests (TFTs), it is important to comment on the fact that she had positive antibodies directed against her thyroid. It cannot be excluded that some of our patient's symptoms, at least in part, might be contributed to thyroid horomone status dysregulation. However, what we have observed are normal TFTs at the time of aura (Table 1.), as well as normal TFTs in both prior (obtained yearly during routine check-ups) and subsequent (obtained twice after the first episode of auras) testing. Positive antithyroid antibodies were observed only during the first episode of aura. In the subsequent test (in September), they were negative. We believe that the anti-thyroid antibodies might have been transiently increased during that time, possibly as a result of a presumed “widespread” autoimmune reaction. It is important to note that autoimmune thyroid disease (especially Graves disease) has been observed both in patients following COVID-19 infection, as well as in several patients following SARS-CoV-2 vaccination

  1. I assume the presentation of this case has no a high scientific impact because of the vague etiologies, and variable symptoms in a wide time period.

We fully agree that no causal connection can be made at this point, which is why we took great care to report this case as adverse events following immunization (AEFI) and to claim no causality: we merely wish to indicate the possibility of the connection. We believe it is important to report this case as a possible connection because it might aid clinicians in case they meet patients with similar presentations and have suspicion that their observations might be related to vaccination, in order explore this potential new avenue of side effects and to drive further research.

  1. In spite of the low scientific impact i agree to publish this case because of interesting sequential symptoms which all could be created from a nerological anatomic site,i.e, NMJ in CNS synapses and NMJ in LMN area.

We thank the Referee for positive response and a possible explanation of pathophysiological link to the condition which is still questionable.

Reviewer 2 Report

Authors reported a case of fasciculations and migraine aura without headache after vaccination against SARS-CoV-2. The present case adds to what is known about that vaccination. However, some points should be clarified. A major flaw of the case report is that the reported symptoms could be triggered by causes other than vaccination.

1 - I suggest specifying whether the woman had a prior personal and/or family history of migraine, with or without aura.

2 - The woman is 48 years old. This is a typical age for menopausal transition, and that age is usually associated with worsening or even onset of migraine with aura.

3 - Authors found signs of thyroid disease in the woman. Fasciculations can be triggered by thyroid disease. Why do Authors think that the trigger of fasciculations was vaccination?

Author Response

We wish to thank Referee 2 for providing us with insightful and important suggestions and remarks, as well as for pointing out the omissions we had made along the way of writing our manuscript. We believe, they truly helped us improve the quality of our manuscript and to improve the reader-experience – to offer future readers interesting and useful information regarding the possible adverse event following immunization with BNT162b2 mRNA vaccine which they may encounter in their clinical practice.

We have rewritten parts of our discussion, as well as updated introduction and case report sections with appropriate information. We used „track changes“ in our manuscript to highlight the changes we made. Furthermore, we stated the most important changes that in our point-by-point response to the Referees (we excluded the reference numbering in the point-by-point response).

Referee: 2

Authors reported a case of fasciculations and migraine aura without headache after vaccination against SARS-CoV-2. The present case adds to what is known about that vaccination. However, some points should be clarified. A major flaw of the case report is that the reported symptoms could be triggered by causes other than vaccination.

We thank the Referee for their insightful comments and interesting remarks. They helped us improve the quality of our paper. We fully agree that no causal connection can be made at this point, which is why we took great care to report this case as adverse events following immunization (AEFI) and to claim no causality: we merely indicate the possibility of the connection.

  1. - I suggest specifying whether the woman had a prior personal and/or family history of migraine, with or without aura.

We thank the Referee for indicating this and we once more highlight that statement in the manuscript. The patient has negative personal of family history of migraine, aura and never experienced any type of headache (she had never taken an analgetic for headache). Aura without headache occurred within a few months after vaccination while muscle fasciculations were still present and that was the first experience of such phenomena.

We added the following sentence in the Case Report section: “The patient had never before experienced migraine-associated phenomena and her family history is also negative for migraine headaches or auras.”

  1. The woman is 48 years old. This is a typical age for menopausal transition, and that age is usually associated with worsening or even onset of migraine with aura.

We thank the Referee for indicating this. We were aware of the above at the time of writing this case. The patient has a negative family history of auras, migraines and headaches and still has regular menstrual cycle (30-32 days), with no signs of cycle shortening and other gynaecological and hormonal observations.

We added this in the very beginning of the Case Report section: “We  present a 48-year-old Caucasian female patient with regular menstrual cycles (30-32 days) suffering from benign fasciculation syndrome that began shortly after recieving the first dose of BNT162b2 mRNA vaccine

  1. Authors found signs of thyroid disease in the woman. Fasciculations can be triggered by thyroid disease. Why do Authors think that the trigger of fasciculations was vaccination?

We thank the Referee for their remarks. The patient had never had fasciculations before vaccination. We report fasciculations as potential adverse events following immunization due to the temporal proximity. We would like to emphasize that we do not claim causality.

Regarding the thyroid, what we have observed are normal thyroid function tests (TFT) at the time of aura (Table 1.), as well as normal TFTs in both prior (obtained yearly during routine check-ups) and subsequent (obtained twice after the first episode of aura) testing. Positive antithyroid antibodies were observed only during the first episode of aura. In the subsequent test, they were negative. We believe they might have been transiently increased during that time. It is also possible that they were increased as a result of a “widespread” autoimmune reaction, which was previously hinted in the discussion. We would also like to mention a few recently published cases which report autoimmune thyroid disease in relation to SARS-CoV-2 vaccination: a) Weintraub et. al, 2021, Journal of Investigative Medicine, High Impact Case Reports, Volume 9: 1–4; b) Vera-Lastra et. al, 2021, ThyroidVol. 31, No. 9; c) Lui et al, 2021, Front. Public Health.

Prompted by the Referee’s interesting remarks, we have rewritten part of the discussion related to their remarks: “Muscle twitching is a known neuromuscular manifestations of COVID-19 infection; migraine headaches and auras have been linked with COVID-19 infection in earlier literature where it was hypothesized that coronaviruses may affect bioelectrical activity of the brain, especially of the occipital lobes. Furthermore, a case of a 38 year-old female patient with known history of migraines that developed status migrainosus one day following the same vaccination has been reported by Consoli et al. We hypothesize that it is possible this might be an acute reaction in a „susceptible“ patient, whereas a migraine aura without headache in our patient 127 days following vaccination might relate to a subacute autoimmune reaction in a patient unburdened with migraine-related issues. It is worthy to note that fasciculations and migraine-related phenomena have been reported as AEFI with other vaccines. To add, a case of subacromial-subdeltoid bursitis following a different COVID-19 (Oxford – AstraZeneca) vaccine has been reported in the medical literature..

Even though, our patient had normal thyroid function tests (TFTs), it is important to comment on the fact that she had positive antibodies directed against her thyroid. It cannot be excluded that some of our patient's symptoms, at least in part, might be contributed to thyroid horomone status dysregulation. However, what we have observed are normal TFTs at the time of aura (Table 1.), as well as normal TFTs in both prior (obtained yearly during routine check-ups) and subsequent (obtained twice after the first episode of auras) testing. Positive antithyroid antibodies were observed only during the first episode of aura. In the subsequent test (in September), they were negative. We believe that the anti-thyroid antibodies might have been transiently increased during that time, possibly as a result of a presumed “widespread” autoimmune reaction. It is important to note that autoimmune thyroid disease (especially Graves disease) has been observed both in patients following COVID-19 infection, as well as in several patients following SARS-CoV-2 vaccination

Round 2

Reviewer 2 Report

I have no further comments.